# Physical Activities in Public Squares: The Impact of Companionship on Chinese Residents' Health



**Xiuhai Xiong [1], Lingbo Liu [1], Zhenghong Peng [2],\* and Hao Wu [2]**

[1] Department of Urban Planning, School of Urban Design, Wuhan University, Wuhan 430072, China; xiongxiuhai@whu.edu.cn (X.X.); lingbo.liu@whu.edu.cn (L.L.)
[2] Department of Graphics and Digital Technology, School of Urban Design, Wuhan University, Wuhan 430072, China; wh79@whu.edu.cn
\* Correspondence: pengzhenghong@whu.edu.cn

**Abstract:** Companionship is the most important social support factor in physical activities, but the influence of companionship on the daily physical activities of Chinese people in the square is not clear. The ordered logistic regression was conducted to identify the companionship and physical activities associated with the physical and mental health of residents ($n = 196$). The results show that companionship has direct and indirect effects on mental health, and companionship acts on physical health through physical activity in public squares. Our research understands the use of public open space (POS) from the perspective of companionship and provides a new perspective for improving the sociality of POS design.

**Keywords:** physical activity; companionship; social support; health status

## 1. Introduction

Evidence has shown that POS provides residents with places for physical exercise and contributes to their physical and mental health, wherein social support is supposed to have a positive effect on increasing physical activity [1]. Social support is defined as providing support or assistance to accomplish a specific behavior [2], including instrumental support, information support, companionship, emotional support, etc. [3,4], wherein companionship is supposed to be one of the most important social support factors in physical activity [5,6]. Companionship can be further divided into the companionship of minors, the companionship of adults, and the companionship of pets [7–9].

With the improvement of health awareness, the physical activity of Chinese residents has also begun to increase in recent years [10]. A large number of residents choose to go to the city square near the residential area for activities, including square dancing, running, and walking. Urban squares are often located near high-density commercial areas and residential areas, and some of them are dominated by hard pavement. Its green coverage is lower than that of parks, and it is also different from common parks abroad. Although there are relatively many studies on companionship, physical activity, and healthy behavior in foreign studies [11,12], the influence of companionship on physical activity and health of Chinese residents in urban squares is not clear, and further exploration is still needed.

Berkman, et al. [13] propose a model for the relationship between social connection and health, suggesting that in health-related behaviors, companionship and physical activity are possible ways to improve health. However, the theoretical and empirical findings of the mechanism behind this hypothesis in China are scarce; this article aims to study the impact of companionship on the health of Chinese residents in the context of physical activities in the city squares.

In previous studies, the effects of companionship on physical activity and health were divided into three aspects: the influence of companionship on physical activities, the influence of companionship on mental health and physical health.

### 1.1. The Impact of Companionship on Physical Activity

Previous studies have shown that companionship has an impact on physical activity, indicating that some promoting factors for physical activity in the process of companionship may provide exercisers with social and emotional support, exercise methods, and exercise fun, and regulate the exerciser's self-efficacy [14,15]. In terms of specific groups, compared with those who exercise alone, companionship is proved to obviously improve the exercise level of women [14–16], minors [17,18], and the elderly [19]. However, the different sources of companionship have various effects on sports, for example, the companionship of friends has a more positive effect on the sports of college students than the companionship of the family [20]. Companionship is, therefore, considered a strategy to increase physical activity and reduce sedentary behavior [16].

### 1.2. The Impact of Companionship on Mental Health

The effect of companionship on mental health could also be divided into direct and indirect influences. Companionship can help improve people's life satisfaction, memory, and prevent depression [21–23], wherein the direct positive impact has been verified existing in the companionship of family, friends, pets, etc., especially for the elder population, and children. Previous studies have shown that elderly people living alone not only have more anxiety and loneliness, but also their cognitive abilities decrease [24,25], which indicates that losing companionship will lead to psychological distress. On the contrary, the companionship of family and friends can alleviate the psychological distress of the elderly [26,27]. As far as children are concerned, only children are more likely to escape difficulties and lack confidence, having a high depression rate [28]. Left-behind children in rural China also tend to be lonely, depressed, and have a lower life satisfaction [9,29], while family members can ease the working pressure of Chinese migrant workers in remote places [30]. In addition, the companionship of pets can not only reduce depression, anxiety, and loneliness, but also increase empathy and socialization of pet owners [31–33].

The indirect impact of companionship in physical activities on mental health is mainly reflected as a mediating variable. In other words, companionship improves the level of physical activity which further enhances mental health. It has been proven that companionship, including the companionship of friends, family members, and pets, has a positive impact on physical activities, active participation in physical activities will further reduce loneliness, improve quality of life, improve mental health, and reduce depression risk [34–40]. People with a high degree of social isolation and low outdoor physical activities may have a very high probability of depression, while groups with friends and neighbors, maintaining high levels of outdoor physical activity, have an extremely low probability of depression [41].

### 1.3. The Impact of Companionship on Physical Health

The impact of companionship on physical health could be divided into direct and indirect impacts. In terms of direct impact, the population which is accompanied by family and friends has better physical health levels than those living alone, such as blood pressure, height, and weight [30]. On the contrary, the elderly who are socially isolated are at risk of physical failure and even death [42,43]. The indirect influence on physical health is mainly reflected as mediating variable in physical activities. Though there are relatively few studies on this topic, related studies showed that pet companionship could stimulate more frequent leisure-time physical activities, which would increase physical activity and significantly enhance the health situation of patients, including obesity, hypertension, and hyperlipidemia [32,33,44,45].

### 1.4. Study Aim and Hypotheses

It must be admitted that companionship is a complicated process with different patterns, such as family [46], friends [9], pets [47], robots [48], social media [49], etc., and there may also be occurring a series of behaviors, such as hugging [50], talking [50], dancing [50],

exercise [50], eating [51], etc. Therefore, in order to explain clearly the role of companionship on health, it is necessary to conduct a comprehensive investigation and research on the pattern and behavior of companionship. Time geography believes that the occurrence of behavior has to be restricted by time and space and the analysis of spatiotemporal behavior is helpful to discover the matching relationship between individual and social systems, so as to find the general social laws behind the behavior [52–54]. As a specific activity space, the squares for physical actives contain different types of companionship and behavior under a certain time and space background, which provides a natural laboratory to study the impact of companionship on the physical and mental health of Chinese residents.

The purpose of the current research is to evaluate the relationship between companionship and physical and mental health, clarify whether this relationship is positive or negative, and further explain the mechanism. Based on questionnaire surveys and interviews with active crowds in the three squares of Hongshan, Shouyi, and Xibeihu in Wuhan, China, the study divided the survey population into a single pattern and companionship pattern which was further divided into minors, adults, and pets. As shown in Figure 1, the hypothesis is listed as follows:

(1) The pattern of companionship directly and positively affects physical and mental health.
(2) Companionship indirectly affects physical and mental health through the intensity of physical exercise.
(3) There is a relationship of mutual influence and mutual promotion between mental and physical health.

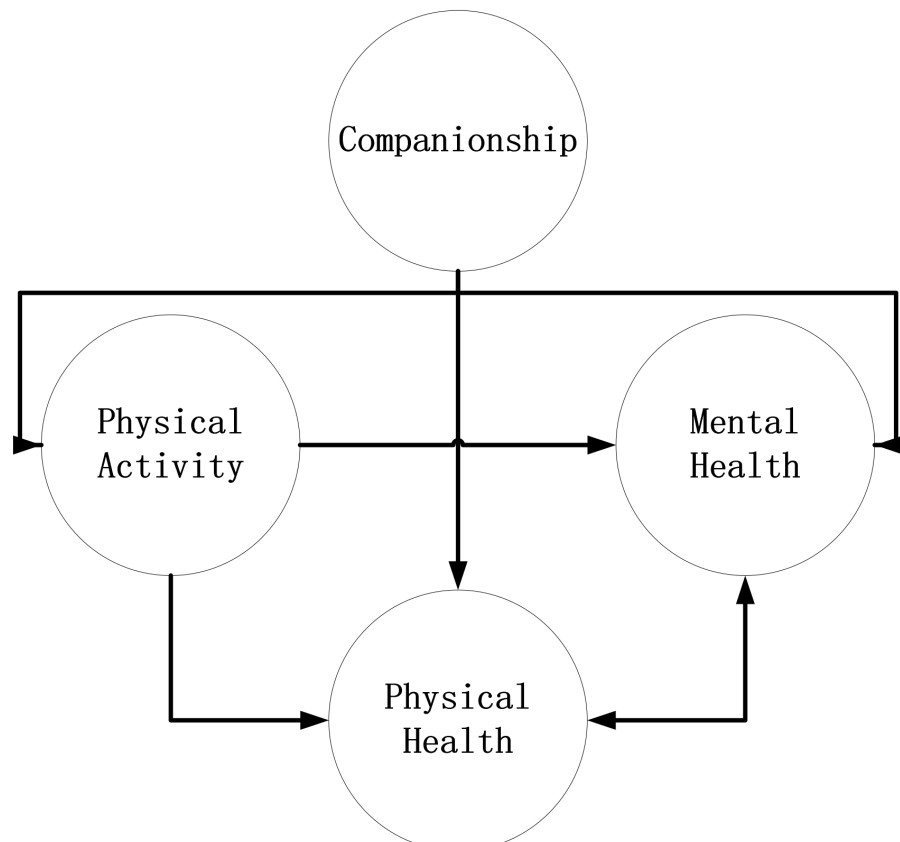

**Figure 1.** Conceptual model with hypothesized paths.

## 2. Materials and Methods

### 2.1. Procedure

The questionnaire-based study was conducted anonymously. As shown in Figure 2, the survey locations are from three squares in the center of Wuhan. There are many squares in Wuhan, but most of the squares are small and belong only to residents of the local community. We excluded these small squares with limited services. We chose a large area where the service area is facing the public square of the whole city. Finally, 3 squares were selected. The three squares are all within the second ring road, surrounded by commercial and residential areas, densely populated, and convenient transportation. The duration of our investigation was from September 2020 to November 2020, which is in the fall. According to the Wuhan Statistical Yearbook, from 2010 to 2019, the average temperature in Wuhan in the spring was 17.11 °C and the average precipitation was 118.87 mm; the average temperature in summer was 27.85 °C and the average precipitation was 191.41 mm; the average temperature in autumn was 17.72°C and the average precipitation was 72.31 mm; the average temperature in winter was 4.97 °C, and the average precipitation in winter was 38.94 mm. Wuhan is rainy in spring, hot in summer, cold in winter, and only sunny and warm in autumn. Therefore, it is especially suitable for residents to carry out physical activities outdoors in Wuhan's autumn. We obtained the weather data during the survey period (1 September 2020 to 1 December 2020). The data comes from the website: https: //rp5.ru/, which can obtain daily historical weather data in Wuhan. Since we chose to investigate on a sunny day and there is no record of precipitation, our average temperature is 18 °C and the average wind speed is 0.48 m/s. The daily survey period lasted from 6 pm to 9 pm. At this time, the workers are off work, the students are home from school, and dinner is over, so most families can get together to participate in sports activities. As mentioned in previous studies, Chinese people are accustomed to going out for exercise after meals and consider this as a health need [50]. The questionnaire survey was conducted face-to-face by the members of the research team; 200 questionnaires were collected, of which 196 were valid questionnaires. The members of the research team clearly explained the whole process, and the subjects voluntarily filled out and responded to the anonymous questionnaire with informed consent. Subjects could suspend or revoke their right to agree to participate in the research at any time without explaining the reason for doing so. All procedures comply with the principles embodied in the "Helsinki Declaration" and were approved by the Ethics Committee of Wuhan University.

### 2.2. Data Collection

#### 2.2.1. Socio-Demographic Data

Socio-demographic data include gender, age, occupation, education status, fitness, sedentary status, travel mode, and travel time from home to the square. As shown in Table 1, there were 79 (40.31%) males and 117 (59.69%) females. In terms of age group, there were 9 (4.59%) people under the age of 18, 84 (42.86%) people in the 19–44 age group, 38 (19.39%) people in the 45–59 age group; there are 65 (33.16%) people in elderly aged 60 years and above group. For *other* instances, see the descriptive statistics in Table 1.

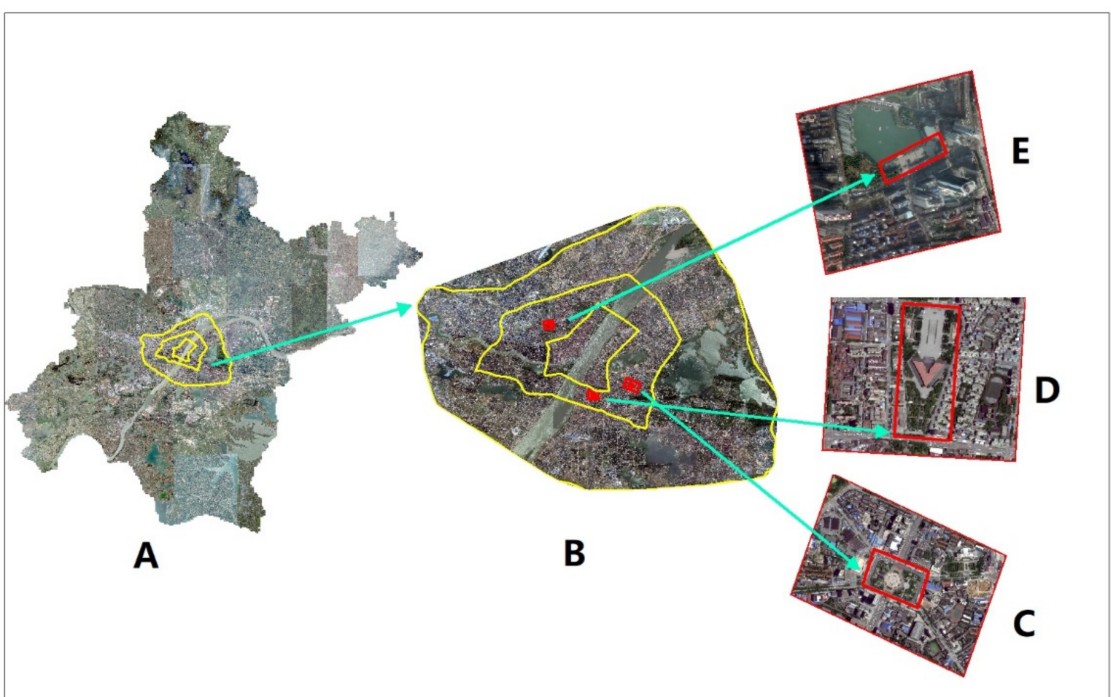

**Figure 2.** The location of the survey site in Wuhan. (**A**) Remote sensing image of Wuhan City; (**B**) remote sensing image map of the area within the third ring road in Wuhan; (**C**) Hongshan Square; (**D**) Shouyi Square; (**E**) Xibeihu Square.

**Table 1.** Socio-economic characteristics of visitors to the square.

| Characteristic | Group | Number | Percentage |
|---|---|---|---|
| Gender | Male | 79 | 40.31% |
| | Female | 117 | 59.69% |
| Age | ≤18 | 9 | 4.59% |
| | 19–44 | 84 | 42.86% |
| | 45–59 | 38 | 19.39% |
| | ≥60 | 65 | 33.16% |
| Occupation | Student | 16 | 8.16% |
| | Employee | 68 | 34.69% |
| | Freelancer | 37 | 18.88% |
| | Retired | 72 | 36.73% |
| | Others | 3 | 1.53% |
| Education | High school and below | 95 | 48.47% |
| | Undergraduate | 86 | 43.88% |
| | Master's degree and above | 15 | 7.65% |
| Fitness frequency | Never go | 137 | 69.90% |
| | Go often | 9 | 4.59% |
| | Go occasionally | 50 | 25.51% |
| Sedentary time | ≤3 | 86 | 43.88% |
| | 3–6 | 64 | 32.65% |
| | 6–9 | 38 | 19.39% |
| | 9–12 | 7 | 3.57% |
| | ≥12 | 1 | 0.51% |

### 2.2.2. Companion Patterns Data

In previous studies, the companionship patterns are mainly divided into the companionship of friends [55], the companionship of family members [55], and the companionship of pets [56]. Our companionship patterns were divided into minor companionship, adult companionship, and pet companionship. This division is more in line with the real situa-

tion of the square. It is worth noting that what we asked the subjects to answer was the daily companionship patterns. As shown in Table 2, in the companionship patterns, 96 (48.98%) people were accompanied by adults, 4 (2.04%) people were accompanied by pets, 56 (28.57%) people were accompanied by minors, and 40 (20.41%) people were alone.

**Table 2.** Descriptive statistics of the companionship patterns.

| Characteristic | Group | Number | Percentage |
| --- | --- | --- | --- |
| | Adult | 96 | 48.98% |
| Companionship | Pet | 4 | 2.04% |
| | Minor | 56 | 28.57% |
| | Alone | 40 | 20.41% |

### 2.2.3. Physical Activity in the Public Squares

As shown in Table 3, the physical activity process in the public square includes transportation mode, travel time, visit frequency, activity type, and physical activity level. The level of physical activity on the square is represented by the stay time and is divided into four levels. The longer the stay, the higher the level of physical activity. The first level is 2 (1.02%) people, the activity time is between 3 and 4 h; the second level is 16 (8.16%) people, the activity time is between 2 and 3 h; the third level is 104 (53.06%) people, the activity time is between 1 and 2 h; the fourth level is 74 (37.76%) people, the activity time is between 0.5 and 1 h. In square activities, 7 (3.57%) people participated in running, 6 people (3.06%) participated in skateboarding, 155 (79.08%) people participated in walking, 26 (13.27%) people participated in dancing, and 22 (11.22%) people participated in others activity. For related content, see the descriptive statistics in Table 3.

**Table 3.** Physical activity in the public squares.

| Characteristic | Group | Number | Percentage |
| --- | --- | --- | --- |
| | Walk | 136 | 69.39% |
| Travel modes | Public transportation | 46 | 23.47% |
| | Cycling | 13 | 6.63% |
| | Others | 1 | 0.51% |
| | 1–5 min | 33 | 16.84% |
| | 6–10 min | 68 | 34.69% |
| Travel time | 11–20 min | 65 | 33.16% |
| | 21–30 min | 20 | 10.20% |
| | ≥30 min | 10 | 5.10% |
| | Times per day | 81 | 41.33% |
| Visit frequency | Several times weekly | 71 | 36.22% |
| | Several times monthly | 27 | 13.78% |
| | Others | 17 | 8.67% |
| | Running | 7 | 3.57% |
| | Skateboarding | 6 | 3.06% |
| Activity type | Walking | 155 | 79.08% |
| | Dancing | 26 | 13.27% |
| | Others | 22 | 11.22% |
| | Very high | 2 | 1.02% |
| Physical activity level | High | 16 | 8.16% |
| | Moderate | 104 | 53.06% |
| | Low | 74 | 37.76% |

### 2.2.4. Data on Physical and Mental Health

Self-reported physical health and self-reported mental health are shown in Table 4. Mental and physical health was obtained through self-reports of visitors to the square. The physical health self-assessment status is divided into 5 levels, "excellent," "good," "fair," "poor," and "bad", corresponding to a score of 1 to 5. The first level represents excellent

health, with a total of 24 (12.24%) people. The second level represents good health, with a total of 83 (42.35%) people. The third level represents fair physical health, and there were 80 (40.82%) people at this level. The fourth level indicates poor physical health, and there were 7 (3.57%) people at this level. The fifth level represents bad physical health, and there were 2 (1.02%) people at this level. Mental health self-assessment is also divided into 5 levels, "excellent," "good," "fair," "poor," and "bad", corresponding to a score of 1 to 5. The first level indicates that the mental health status is excellent, and there were 43 (21.94%) people at this level. The second level indicates that the mental health status is good, and there were 99 (50.51%) people at this level. The third level indicates that the mental health status of the population is fair, and there were 50 (25.51%) people at this level. The fourth level indicates that the mental health of the population is poor, and there were 4 (2.04%) people at this level. The fifth level is bad mental health, and there is no distribution in this level.

**Table 4.** Descriptive characteristics of respondents' mental health and physical health.

| Characteristic | Group | Number | Percentage |
|---|---|---|---|
| | Excellent | 24 | 12.24% |
| | Good | 83 | 42.35% |
| Physical health | Fair | 80 | 40.82% |
| | Poor | 7 | 3.57% |
| | Bad | 2 | 1.02% |
| | Excellent | 43 | 21.94% |
| | Good | 99 | 50.51% |
| Mental health | Fair | 50 | 25.51% |
| | Poor | 4 | 2.04% |

## 3. Statistical Analysis

In the process of statistical analysis, we used the analysis method of ordered logistic regression, and this method was used three times. For the first time, we used demographic variables, different types of companionship and types of physical activity as covariates, and the level of physical activity as dependent variables. For the second time, we used demographic variables, different types of companionship and types of physical activity, the physical activity level, and mental health self-assessment scores as covariates, and self-assessed physical health as dependent variables. For the third time, we used demographic variables, different types of companionship, and types of physical activity. The physical activity level and physical health score were used as covariates, and self-evaluated mental health scores were used as dependent variables. The statistical analysis platform used SPSS Statistics 25.0 (IBM SPSS Statistics, New York, NY, USA).

## 4. Result

### 4.1. Influencing Factors of Physical Activity

To find the influencing factors of the physical activity level, this study selected 33 variables from social-demographic indicators, companionship pattern indicators, and square physical activity indicators as covariates for an ordered logistic regression analysis (Table 5). The results show that the model had a good fitting and significance (Nagelkerke's pseudo $R^2 = 0.413$, $\chi^2 = 85.092$, df = 30, $p < 0.001$), and the parallel line test was not significant ($\chi^2 = 72.060$, df = 60, $p = 0.136$). First of all, in the companionship pattern, the companionship of pets (OR = 797.12), companionship of minors (OR = 7.43), and companionship of adults (OR = 2.44) showed a positive effect on increasing the level of physical activity. Secondly, sedentary time $\leq 3$ (OR = 0.01), 3–6 h (OR = 0.01), 6–9 h (OR = 0.01) and 9–12 h (OR = 0.01) had a negative impact on increasing physical activity. Third, the frequency of daily visits (OR = 2.94) had a positive effect on the increase in physical activity levels. Fourth, the activity type dancing (OR = 2.78) had a positive effect on the improvement of

physical activity level. Fifth, in the square type, the Northwest Lake Square (OR = 2.27) had a positive effect on the improvement of the physical activity level.

**Table 5.** Ordinal logistic regression analysis on the influencing factors of physical activity level.

| Characteristic | Variable | Estimate | Exp(B) | 95%CI | | *p*-Value |
|---|---|---|---|---|---|---|
| Gender | Male (ref) | | | | | |
| | Female | 0.39 | 1.47 | 0.72 | 3.00 | ns |
| Age | ≥60 (ref) | | | | | |
| | ≤18 | −1.82 | 0.16 | 0.01 | 3.33 | ns |
| | 19–44 | 0.37 | 1.44 | 0.30 | 7.02 | ns |
| | 45–59 | 0.46 | 1.59 | 0.39 | 6.5 | ns |
| Occupation | Retired (ref) | | | | | |
| | Fixed occupation | −0.21 | 0.81 | 0.17 | 3.87 | ns |
| | Student | 1.06 | 2.87 | 0.29 | 27.97 | ns |
| | Freelancer | −0.13 | 0.88 | 0.19 | 4.02 | ns |
| Education | High school and below (ref) | | | | | |
| | Undergraduate | −0.39 | 0.68 | 0.29 | 1.58 | ns |
| | Master's degree and above | 0.33 | 1.39 | 0.28 | 6.83 | ns |
| Fitness frequency | Never go (ref) | | | | | |
| | Go occasionally | 0.19 | 1.21 | 0.56 | 2.62 | ns |
| | Go often | −0.76 | 0.47 | 0.09 | 2.53 | ns |
| Sedentary time | ≥12 (ref) | | | | | |
| | ≤3 h | −4.57 | 0.01 | 0.00 | 1.44 | * |
| | 3–6 h | −4.32 | 0.01 | 0.00 | 1.90 | * |
| | 6–9 h | −4.99 | 0.01 | 0.00 | 1.00 | ** |
| | 9–12 h | −5.20 | 0.01 | 0.00 | 0.94 | ** |
| Travel time | ≥30 min (ref) | | | | | |
| | 1–5 min | 0.52 | 1.69 | 0.27 | 10.40 | ns |
| | 6–10 min | 1.27 | 3.56 | 0.69 | 18.41 | ns |
| | 11–20 min | 1.11 | 3.02 | 0.57 | 15.86 | ns |
| | 21–30 min | 0.63 | 1.87 | 0.30 | 11.81 | ns |
| Companionship | Alone (ref) | | | | | |
| | Pet | 6.68 | 797.12 | 47.99 | 13,240.03 | *** |
| | Minor | 2.01 | 7.43 | 2.75 | 20.09 | *** |
| | Adult | 0.89 | 2.44 | 0.98 | 6.10 | * |
| Visit frequency | Several times monthly (ref) | | | | | |
| | Times per day | 1.08 | 2.94 | 1.08 | 7.99 | ** |
| | Several times weekly | 0.08 | 1.08 | 0.44 | 2.65 | ns |
| Activity type | Skateboarding (ref) | | | | | |
| | Walking | −0.40 | 0.67 | 0.27 | 1.68 | ns |
| | Running | −0.75 | 0.47 | 0.08 | 2.97 | ns |
| | Dancing | 1.02 | 2.78 | 0.87 | 8.89 | * |
| Square name | Shouyi Square (ref) | | | | | |
| | Hongshan Square | 0.32 | 1.38 | 0.65 | 2.96 | ns |
| | Xibeihu Square | 0.82 | 2.27 | 0.92 | 5.63 | * |

Note: The significant expressed as (* *p* < 0.1), (** *p* < 0.05), (*** *p* < 0.001), and ns (no significance).

### 4.2. Influencing Factors of Mental Health

In order to find the influencing factors of mental health, this study selected 36 variables from socio-demographic indicators, companionship pattern indicators, physical health indicators, and square physical activity indicators as covariates for the ordered logistic regression analysis (Table 6). The results show that the model had a good fit and significance (Nagelkerke's pseudo $R^2$ = 0.413, $\chi^2$ = 85.092, df = 30, $p$ < 0.001), and the parallel line test was not significant ($\chi^2$ = 72.060, df = 60, $p$ = 0.136). First, the companionship of pets (OR = 69.20), companionship of minors (OR = 3.49), and companionship of adults (OR = 3.01) showed positive effects on the improvement of mental health. Second, excellent (OR = 41,982.16), good (OR = 2268.79), fair (OR = 280.90), and poor (OR = 206.23) physical health had a positive effect on the improvement of mental health. Third, among the ages, 19–44 years

(OR = 0.21) showed a negative effect on the improvement of mental health. Fourth, the visit frequency per day (OR = 3.00) showed a positive effect on the improvement of mental health. Fifth, extremely high (OR = 0.02) and average (OR = 0.42) levels of physical activity showed a negative effect on the improvement of mental health.

**Table 6.** Ordered logistic regression analysis on the influencing factors of mental health.

| Characteristic | Variable | Estimate | Exp(B) | 95%CI | | *p*-Value |
|---|---|---|---|---|---|---|
| Gender | Male (ref) | | | | | |
| | Female | −0.04 | 0.96 | 0.48 | 1.93 | ns |
| Age | ≥60 (ref) | | | | | |
| | ≤18 | −1.67 | 0.19 | 0.01 | 3.59 | ns |
| | 19–44 | −1.54 | 0.21 | 0.05 | 1.00 | ** |
| | 45–59 | −0.96 | 0.38 | 0.10 | 1.50 | ns |
| Occupation | Retired (ref) | | | | | |
| | Fixed occupation | 0.67 | 1.95 | 0.44 | 8.59 | ns |
| | Student | 1.31 | 3.71 | 0.35 | 39.06 | ns |
| | Freelancer | 0.14 | 1.15 | 0.27 | 4.91 | ns |
| Education | High school and below (ref) | | | | | |
| | Undergraduate | 0.03 | 1.03 | 0.46 | 2.32 | ns |
| | Master's degree and above | 0.69 | 1.99 | 0.42 | 9.39 | ns |
| Fitness frequency | Never go (ref) | | | | | |
| | Go occasionally | 0.20 | 1.23 | 0.58 | 2.61 | ns |
| | Go often | 1.22 | 3.39 | 0.52 | 22.13 | ns |
| Companionship | Alone (ref) | | | | | |
| | Pet | 4.24 | 69.20 | 1.71 | 2801.75 | ** |
| | Minor | 1.25 | 3.49 | 1.27 | 9.57 | ** |
| | Adult | 1.10 | 3.01 | 1.24 | 7.28 | ** |
| Visit frequency | Several times monthly (ref) | | | | | |
| | Times per day | 1.10 | 3.00 | 1.12 | 8.03 | ** |
| | Several times weekly | 0.23 | 1.26 | 0.54 | 2.97 | ns |
| Activity type | Skateboarding (ref) | | | | | |
| | Walking | −0.90 | 0.41 | 0.16 | 1.04 | * |
| | Running | 0.95 | 2.58 | 0.45 | 14.61 | ns |
| | Dancing | 0.12 | 1.13 | 0.35 | 3.65 | ns |
| Physical health | Extremely poor (ref) | | | | | |
| | Excellent | 10.65 | 41,982.16 | 1215.61 | 1,451,343.16 | *** |
| | Good | 7.73 | 2268.79 | 85.97 | 59,874.14 | *** |
| | Fair | 5.64 | 280.90 | 11.78 | 6707.62 | *** |
| | Poor | 5.33 | 206.23 | 5.48 | 7769.80 | ** |
| Physical activity level | Low (ref) | | | | | |
| | Very high | −3.74 | 0.02 | 0.00 | 0.82 | ** |
| | High | −0.69 | 0.50 | 0.12 | 2.13 | ns |
| | Moderate | −0.87 | 0.42 | 0.20 | 0.90 | ** |

Note: The significant expressed as (* $p < 0.1$), (** $p < 0.05$), (*** $p < 0.001$), and ns (no significance).

### 4.3. Influencing Factors of Physical Health

To find the influencing factors of physical health, this study selected 40 variables from social-demographic indicators, companionship pattern indicators, mental health indicators, and square physical activity indicators as covariates for the ordered logistic regression analysis (Table 7). The results show that the model had good fit and significance (Nagelkerke's pseudo $R^2$ = 0.536, $\chi^2$ = 129.112, df = 30, $p < 0.001$) parallel line test was not significant ($\chi^2$ = 29.828, df = 90, $p$ = 1) First, the companionship of minors in companionship (OR = 0.27) showed a negative effect on the improvement of physical health. Second, the general (OR = 9.23) and high (OR = 3.73) levels of physical activity showed a positive effect on the improvement of physical health. Third, in mental health, general (OR = 6.83), good (OR = 52.98), and excellent (OR = 716.23) showed a positive effect on the improvement of physical health. Fourth, the reduction in sedentary time ≤ 3 (OR = 6843.13), 3–6 h (OR = 3540.42), 6–9 h (OR = 5146.13), 9–12 h (OR = 2494.89) showed improvement in the

physical health impact. Fifth, in the type of physical activity, dancing (OR = 0.29) showed a negative effect on the improvement of physical health.

**Table 7.** Orderly logistic regression analysis of influencing factors of physical health.

| Characteristic | Variable | Estimate | Exp(B) | 95%CI | | *p*-Value |
|---|---|---|---|---|---|---|
| Gender | Male (ref) | | | | | |
| | Female | 0.11 | 1.12 | 0.56 | 2.24 | ns |
| Age | ≥60 (ref) | | | | | |
| | ≤18 | 1.80 | 6.06 | 0.31 | 117.10 | ns |
| | 19–44 | 0.61 | 1.84 | 0.39 | 8.59 | ns |
| | 45–59 | 0.06 | 1.06 | 0.27 | 4.17 | ns |
| Occupation | Retired (ref) | | | | | |
| | Fixed occupation | 0.46 | 1.59 | 0.36 | 7.07 | ns |
| | Student | 1.03 | 2.81 | 0.30 | 26.44 | ns |
| | Freelancer | 0.91 | 2.49 | 0.58 | 10.83 | ns |
| Education | High school and below (ref) | | | | | |
| | Undergraduate | | | | | ns |
| | Master's degree and above | 0.09 | 1.09 | 0.22 | 5.30 | ns |
| Fitness frequency | Never go (ref) | | | | | |
| | Go occasionally | −0.29 | 0.75 | 0.35 | 1.60 | ns |
| | Go often | 0.77 | 2.16 | 0.43 | 10.97 | ns |
| Companionship | Alone (ref) | | | | | |
| | Pet | −2.09 | 0.12 | 0.01 | 2.31 | ns |
| | Minor | −1.32 | 0.27 | 0.10 | 0.72 | ** |
| | Adult | −0.68 | 0.51 | 0.21 | 1.23 | ns |
| Visit frequency | Several times monthly (ref) | | | | | |
| | Times per day | −0.62 | 0.54 | 0.20 | 1.43 | ns |
| | Several times weekly | −0.09 | 0.92 | 0.39 | 2.18 | ns |
| Activity type | Cycling (ref) | | | | | |
| | Walking | −0.32 | 0.73 | 0.29 | 1.86 | ns |
| | Running | −0.67 | 0.51 | 0.09 | 2.86 | ns |
| | Dancing | −1.24 | 0.29 | 0.09 | 0.91 | ** |
| Physical activity level | Low (ref) | | | | | |
| | Very high | −0.10 | 0.90 | 0.02 | 38.86 | ns |
| | High | 2.22 | 9.23 | 1.98 | 43.12 | ** |
| | Moderate | 1.32 | 3.73 | 1.74 | 7.97 | ** |
| Mental health | Poor (ref) | | | | | |
| | Excellent | 6.57 | 716.23 | 61.44 | 8341.51 | *** |
| | Good | 3.97 | 52.98 | 5.34 | 525.32 | ** |
| | Fair | 1.92 | 6.83 | 0.71 | 65.50 | * |
| Sedentary time | ≥12 (ref) | | | | | |
| | 3≤ | 8.83 | 6843.13 | 64.52 | 725,778.39 | *** |
| | 3–6 h | 8.17 | 3540.42 | 33.75 | 371,758.60 | ** |
| | 6–9 h | 8.55 | 5146.13 | 48.86 | 542,530.73 | *** |
| | 9–12 h | 7.82 | 2494.89 | 20.99 | 296,262.15 | ** |

Note: The significant expressed as (* $p < 0.1$), (** $p < 0.05$), (*** $p < 0.001$), and ns (no significance).

## 5. Discussion

### 5.1. The Direct and Indirect Effects of Companionship on Mental Health

On the one hand, this study shows that increasing daily companionship has a direct and positive impact on improving mental health. Previous studies have shown that the companionship of friends and pets directly affects the mental health of the elderly [32,57], which is similar to our research results. On the other hand, companionship improves physical health through physical activity, which indirectly contributes to the improvement of mental health. Previous studies have already shown evidence that the companionship of pet dogs, the companionship of friends, and the companionship of family members improve physical health through physical activities so that mental health is improved [58–60].

### 5.2. The Indirect Impact of Companionship on Physical Health

The indirect impact of companionship on physical health is reflected in two aspects. First of all, this study shows that increasing daily companionship indirectly affects physical health by increasing the level of physical activity. In addition, this study shows that mental health acts as an indirect factor of companionship and physical health, and companionship improves mental health, which will improve physical health. Previous research believed that the higher the activity level, the better the body function [61], which is similar to our research results. There are many reasons for the influence of companionship on the level of physical activity. In the case of females, companionship gives them a sense of security during night activities because they worry about crime, violence, and harassment at night [50]. In addition, emotional support and information feedback in daily companionship are positive factors for tourists to improve their physical exercise level [1]. What's more, our research results also show that sedentary behavior is extremely harmful to the body. Previous studies have shown that sedentary behavior is strongly correlated with obesity, sexual and reproductive diseases [62,63]. Therefore, this study provides strategies to improve physical activity, reduce sedentary behavior, and improve health through companionship. Another factor affecting physical health is mental health. In this regard, previous studies have shown that the relationship between physical and mental health is interdependent and mutually reinforcing [64,65]. This study enriches the relationship between physical and mental health from the perspective of companionship.

### 5.3. Some Practical Implications for POS Design

We believe that two points should be given priority in the design of POS: In the first place, the accessibility and convenience of POS travel. Our survey respondents were mostly middle-aged and elderly people (53%), who mainly choose to travel by walking (69%). In the context of healthy aging, we must pay attention to the accessibility of the elderly to POS and the comfort of the walking environment. Moreover, we must pay attention to social elements in the design of POS, and consider the needs of companionship. For example, children like a flat and open POS where they can run and play [50]. The square dance activities of the elderly take up more space and need bright lights at night [50]. Meantime, some elderly people's rest areas require a lot of seats. Couples like quiet places with dense vegetation [50]. Finally, previous studies have shown that in addition to China, there are also different patterns of companionship in POS research in other countries [15,62], such as the companionship of family and friends, and the companionship of pets. Therefore, the method of understanding and designing POS from the perspective of companionship should be extended to POS of other cultures (countries).

## 6. Conclusions

In this study, three squares in Wuhan, China were used to test the role of companionship in public squares in improving physical activity and promoting physical and mental health. First of all, the influence of companionship on mental health was divided into direct influence and indirect influence. On the one hand, companionship has a direct and positive impact on good mental health. On the other hand, companionship improves physical health through physical activity, which indirectly affects the improvement of mental health. Furthermore, there are two paths for the indirect impact of companionship on physical health. One path indicates that companionship can improve physical health by increasing physical activity levels, and the other path indicates that companionship can achieve better physical health by improving mental health.

The advantage of this study is that it is the first to propose to understand the use of POS in China from the perspective of companionship. The square was used as a natural laboratory to evaluate the daily companionship and physical and mental health of visitors to the square. Meantime, this research further explored the method of designing POS from the perspective of companionship, which can be extended to POS of other cultures (countries).

This study has some limitations. In future research, more attention should be paid to the accuracy and diversity of indicators in terms of health measurement. For example, in the measurement, attention must be paid to the combination of the subject's self-evaluation and the monitoring data of the health equipment. In addition, this study adopts a cross-sectional study, and future studies should have long-term follow-up longitudinal studies.

**Author Contributions:** Conceptualization, X.X. and L.L.; methodology, X.X. and L.L.; investigation, X.X.; software, X.X. and H.W. and X.X.; validation, H.W., L.L.; data curation, X.X.; writing—original draft preparation, X.X.; writing—review and editing, X.X., L.L., Z.P.; supervision, Z.P., L.L. and H.W. All authors have read and agreed to the published version of the manuscript.

**Funding:** This research was funded by National Natural Science Foundation of China (No. 51978535); Humanities and Social Science Project of the Ministry of Education (No. 19YJCZH187).

**Informed Consent Statement:** Informed consent was obtained from all subjects involved in the study.

**Data Availability Statement:** Data sharing not applicable.

**Conflicts of Interest:** The authors declare no conflict of interest.

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
