# Peer review of "Physical Activities in Public Squares: The Impact of Companionship on Chinese Residents’ Health"

_land, doi:10.3390/land10070720_

Round 1
Reviewer 1 Report
While the manuscript is more or less Ok in the present version, I have two issues that should be considered before publication.
- To me, the conclusion section is too long and not focused on the main topic. I would strongly encourage authors to rebalance discussion and conclusions, providing a closing statement in the conclusive chapter.
- Language usage is still heterogeneous throughout the ms, and I would encourage a thorough revision of longer sentences. I believe most of them (e.g. discussion) can be easily rephrased addying value and clarity.
- Thank you for giving me the possibility to read this complete paper.
Author Response
Dear Reviewer:
Thank you for your letter and for the reviewer’s comments concerning our manuscript entitled “Physical Activities in Public Squares: the Impact of Companionship on Chinese Residents' Health” (ID: 1270633). Those comments are all valuable and very helpful for revising and improving our paper, as well as the important guiding significance to our researches. We have studied comments carefully and have made a correction which we hope meet with approval. The main corrections in the paper and the responses to the reviewer’s comments are as flowing:
Point 1:To me, the conclusion section is too long and not focused on the main topic. I would strongly encourage authors to rebalance discussion and conclusions, providing a closing statement in the conclusive chapter.
Response 1:We have rewritten the conclusion and discussion to make the conclusion and discussion clearer and concise, and more closely related to the main theme. In the previous version, the discussion part did not focus on the main results, but extensively discussed sports, physical health, and mental health. We modified this discussion method without a central goal. In the revised version, we mainly focused on Three key results are discussed, which are: 5.1 The direct and indirect effects of companionship on mental health, 5.2 The indirect impact of companionship on physical health, and 5.3 Some practical implications for POS design. In the conclusion of the article, the previous version was lengthy and complicated and did not highlight the main thrust of the article. In the revised version, the content of the 6 paragraphs in the conclusion section has been simplified into three paragraphs. The sentences are simpler and more accurate. We believe that the discussion part and the conclusion part of the revised version are closely related to serve the main topic. The following is the revised relevant part of the paper.
5 Discussion
5.1 The direct and indirect effects of companionship on mental health
On the one hand, this study shows that increasing daily companionship has a direct and positive impact on improving mental health. Previous studies have shown that the companionship of friends and pets directly affects the mental health of the elderly[61,62], which is similar to our research results. On the other hand, companionship improves physical health through physical activity, which indirectly contributes to the improvement of mental health. Previous studies have already shown evidence that the companionship of pet dogs, the companionship of friends, and the companionship of family members improve physical health through physical activities so that mental health is improved[63-65].
5.2 The indirect impact of companionship on physical health
The indirect impact of companionship on physical health is reflected in two aspects. First of all, this study shows that increasing daily companionship indirectly affects physical health by increasing the level of physical activity. In addition, this study shows that mental health acts as an indirect factor of companionship and physical health, and companionship improves mental health, which will improve physical health. Previous research believed that the higher the activity level, the better the body function [66], which is similar to our research results. There are many reasons for the influence of companionship on the level of physical activity. In the case of females, companionship gives people a sense of security during night activities because they worry about crime, violence, and harassment at night[54]. In addition, emotional support and information feedback in daily companionship are positive factors for tourists to improve their physical exercise level [1]. What’s more, our research results also show that sedentary behavior is extremely harmful to the body. Previous studies have shown that sedentary behavior is strongly correlated with obesity, sexual and reproductive diseases[67,68]. Therefore, this study provides strategies to improve physical activity, reduce sedentary behavior, and improve health through companionship. Another factor affecting physical health is mental health. In this regard, previous studies have shown that the relationship between physical and mental health is interdependent and mutually reinforcing[69,70]. This study enriches the relationship between physical and mental health from the perspective of companionship.
5.3 Some practical implications for POS design
We believe that two points should be given priority in the design of POS: In the first place, the accessibility and convenience of POS travel. Our survey respondents are mostly middle-aged and elderly people (53%), who mainly choose to travel by walk (69%). In the context of healthy aging, we must pay attention to the accessibility of the elderly to the POS and the comfort of the walking environment. Moreover, we must pay attention to social elements in the design of POS, consider the needs of companionship. For example, children like a flat and open POS where they can run and play[54]. The square dance activities of the elderly take up more space and need bright lights at night[54]. Meantime, some elderly people’s rest areas require a lot of seats. Couples like quiet places with dense vegetation[54]. Finally, previous studies have shown that in addition to China, there are also different patterns of companionship in POS research in other countries[15,62,71],such as the companionship of family and friends, and the companionship of pets. Therefore, the method of understanding and designing POS from the perspective of companionship should be extended to POS of other cultures (countries).
6 Conclusion
In this study, three squares in Wuhan, China were used to test the role of companionship in public squares in improving physical activity and promoting physical and mental health. First of all, the influence of companionship on mental health is divided into direct influence and indirect influence. On the one hand, companionship has a direct and positive impact on good mental health. On the other hand, companionship improves physical health through physical activity, which indirectly affects the improvement of mental health. Furthermore, there are two paths for the indirect impact of companionship on physical health. One path indicates that companionship can improve physical health by increasing physical activity levels, and the other path indicates that companionship can achieve better physical health by improving mental health.
The advantage of this study is that it is the first to propose to understand the use of POS in China from the perspective of companionship. The square is used as a natural laboratory to evaluate the daily companionship and physical and mental health of visitors to the square. Meantime, this research further explores the method of designing POS from the perspective of companionship, which can be extended to POS of other cultures (countries).
This study has some limitations. In future research, more attention should be paid to the accuracy and diversity of indicators in terms of health measurement. For example, in the measurement, attention must be paid to the combination of the subject's self-evaluation and the monitoring data of the health equipment. In addition, this study adopts a cross-sectional study, and future studies should have long-term follow-up longitudinal studies.
Point 2:Language usage is still heterogeneous throughout the ms, and I would encourage a thorough revision of longer sentences. I believe most of them (e.g. discussion) can be easily rephrased adding value and clarity.
Response 2: We checked and revised the long sentences throughout the article, and finally made these expressions simple, direct, and accurate.
We appreciate for Reviewer’s warm work earnestly and hope that the correction will meet with approval.
Once again, thank you very much for your comments and suggestions.

Reviewer 2 Report
The authors have chosen a very interesting subject of elderly people, public areas, and companionship. The case studies include 3 public squares in Wuhan. Whereas the analysis looks OK, I am still missing the info concerning the methodology - i.e. why the authors chose particular squares? I am sure that there are many other public squares in Wuhan, so please explain what the key was?
Starting from line 130 - the authors explain the methodology, which needs to be more precise. 1. what are the exact climatic parameters in Wuhan 2. the weather conditions under which the survey was conducted 3. what does random choice mean there were several groups of people present and how was this random choice decided on.
The reviewer did not find any discussion on the issue of accessibility for elderly people. Planning and design of open spaces tend to stress the physical and safety needs of the elderly, while social needs are not frequently addressed. From the reviewers' point of view, these issues should be discussed simultaneously - at least in the discussion sector.
There is some similar research in other countries outside China - comparison would be of benefit to the paper.
Author Response
Dear Reviewer:
Thank you for your letter and for the reviewer’s comments concerning our manuscript entitled “Physical Activities in Public Squares: the Impact of Companionship on Chinese Residents' Health” (ID: 1270633). Those comments are all valuable and very helpful for revising and improving our paper, as well as the important guiding significance to our researches. We have studied comments carefully and have made a correction which we hope meet with approval. The main corrections in the paper and the responses to the reviewer’s comments are as flowing:
Point 1:The authors have chosen a very interesting subject of elderly people, public areas, and companionship. The case studies include 3 public squares in Wuhan. Whereas the analysis looks OK, I am still missing the info concerning the methodology - i.e. why the authors chose particular squares? I am sure that there are many other public squares in Wuhan, so please explain what the key was?
Response 1:In order to clearly explain the reasons for choosing the three squares, we have added relevant descriptions in the article. Specifically, there are many squares in Wuhan, but most of the squares are small and belong only to residents of the local community. We exclude these small squares with limited services. Choose a large area and the service area is facing the public square of the whole city. Finally, 3 squares were selected. The following is the revised relevant part of the paper.
2.1. Procedure
The questionnaire-based study was conducted anonymously. As shown in Figure 2, the survey locations are from three squares in the center of Wuhan. There are many squares in Wuhan, but most of the squares are small and belong only to residents of the local community. We exclude these small squares with limited services. Choose a large area and the service area is facing the public square of the whole city. Finally, 3 squares were selected. The three squares are all within the second ring road, surrounded by commercial and residential areas, densely populated, and convenient transportation. The duration of our investigation is from September 2020 to November 2020, which is in the fall. According to the Wuhan Statistical Yearbook, from 2010 to 2019, the average temperature in Wuhan in spring was 17.11°C and the average precipitation was 118.87 mm; the average temperature in summer was 27.85°C and the average precipitation was 191.41 mm; the average temperature in autumn was It is 17.72°C and the average precipitation is 72.31 mm; the average temperature in winter is 4.97°C, and the average precipitation in winter is 38.94 mm. Wuhan is rainy in spring, hot in summer, cold in winter, and only sunny and warm in autumn. Therefore, it is especially suitable for residents to carry out physical activities outdoors in Wuhan’s autumn. The weather reference website during the investigation: https://rp5.ru/, which can obtain daily historical weather data in Wuhan. Since we chose to investigate on a sunny day and there is no record of precipitation, our average temperature is 18℃ and the average wind speed is 0.48m/s. The daily survey period lasts from 6 pm to 9 pm. At this time, the workers are off work, the students are home from school, and dinner is over, so most families can get together to participate in sports activities. As mentioned in previous studies, Chinese people are accustomed to going out for exercise after meals and consider this as a health need [54]. The questionnaire survey was conducted face-to-face by the members of the research team; 200 questionnaires were collected, of which 196 were valid questionnaires. The members of the research team will clearly explain the whole process, and the subjects voluntarily fill out and respond to the anonymous questionnaire with informed consent. Subjects can suspend or revoke their right to agree to participate in the research at any time without explaining the reason for doing so. All procedures comply with the principles embodied in the "Helsinki Declaration" and have been approved by the Ethics Committee of Wuhan University.
Point 2:Starting from line 130 - the authors explain the methodology, which needs to be more precise. 1. what are the exact climatic parameters in Wuhan 2. the weather conditions under which the survey was conducted 3. what does random choice mean there were several groups of people present and how was this random choice decided on.
Response 2:In order to describe the method of the article more accurately. 1.We obtained climate data from 2010 to 2019 from the "Wuhan Statistical Yearbook", which includes monthly average temperature and precipitation. On this basis, we calculated the average distribution of temperature and precipitation in spring, summer, autumn, and winter over the past ten years. The results show that the average temperature in spring in Wuhan is 17.11°C and the average precipitation is 118.87 mm; the average temperature in summer is 27.85°C and the average precipitation is 191.41 mm; the average temperature in autumn is 17.72°C and the average precipitation is 72.31 mm; the average in winter The temperature is 4.97°C and the average precipitation is 38.94 mm. 2. The weather reference website during the investigation: https://rp5.ru/, which can obtain daily historical weather data in Wuhan. Since we chose to investigate on a sunny day and there is no record of precipitation, our average temperature is 18℃ and the average wind speed is 0.48m/s. 3. After reviewing the literature, the expression "random selection" in the method description was not appropriate. We revised this sentence to The questionnaire survey was conducted face-to-face by the members of the research team; 200 questionnaires were collected, of which 196 were valid questionnaires. The expression draws on a previously published article, which is titled: Physical activity areas in urban parks and their use by the elderly from two cities in China and Germany. The following is the revised relevant part of the paper.
2.1. Procedure
The questionnaire-based study was conducted anonymously. As shown in Figure 2, the survey locations are from three squares in the center of Wuhan. There are many squares in Wuhan, but most of the squares are small and belong only to residents of the local community. We exclude these small squares with limited services. Choose a large area and the service area is facing the public square of the whole city. Finally, 3 squares were selected. The three squares are all within the second ring road, surrounded by commercial and residential areas, densely populated, and convenient transportation. The duration of our investigation is from September 2020 to November 2020, which is in the fall. According to the Wuhan Statistical Yearbook, from 2010 to 2019, the average temperature in Wuhan in spring was 17.11°C and the average precipitation was 118.87 mm; the average temperature in summer was 27.85°C and the average precipitation was 191.41 mm; the average temperature in autumn was It is 17.72°C and the average precipitation is 72.31 mm; the average temperature in winter is 4.97°C, and the average precipitation in winter is 38.94 mm. Wuhan is rainy in spring, hot in summer, cold in winter, and only sunny and warm in autumn. Therefore, it is especially suitable for residents to carry out physical activities outdoors in Wuhan’s autumn. The weather reference website during the investigation: https://rp5.ru/, which can obtain daily historical weather data in Wuhan. Since we chose to investigate on a sunny day and there is no record of precipitation, our average temperature is 18℃ and the average wind speed is 0.48m/s. The daily survey period lasts from 6 pm to 9 pm. At this time, the workers are off work, the students are home from school, and dinner is over, so most families can get together to participate in sports activities. As mentioned in previous studies, Chinese people are accustomed to going out for exercise after meals and consider this as a health need [54]. The questionnaire survey was conducted face-to-face by the members of the research team; 200 questionnaires were collected, of which 196 were valid questionnaires. The members of the research team will clearly explain the whole process, and the subjects voluntarily fill out and respond to the anonymous questionnaire with informed consent. Subjects can suspend or revoke their right to agree to participate in the research at any time without explaining the reason for doing so. All procedures comply with the principles embodied in the "Helsinki Declaration" and have been approved by the Ethics Committee of Wuhan University.
Point 3:The reviewer did not find any discussion on the issue of accessibility for elderly people. Planning and design of open spaces tend to stress the physical and safety needs of the elderly, while social needs are not frequently addressed. From the reviewers' point of view, these issues should be discussed simultaneously - at least in the discussion sector.
Response 3:We rewritten the discussion part. We discussed the physical activities of the elderly in public spaces from the perspectives of walking environment, accessibility, and companionship, and discussed the impact of elderly activities on the design of public open spaces. The following is the revised relevant part of the paper.
5.3 Some practical implications for POS design
We believe that two points should be given priority in the design of POS: In the first place, the accessibility and convenience of POS travel. Our survey respondents are mostly middle-aged and elderly people (53%), who mainly choose to travel by walk (69%). In the context of healthy aging, we must pay attention to the accessibility of the elderly to the POS and the comfort of the walking environment. Moreover, we must pay attention to social elements in the design of POS, consider the needs of companionship. For example, children like a flat and open POS where they can run and play[54]. The square dance activities of the elderly take up more space and need bright lights at night[54]. Meantime, some elderly people’s rest areas require a lot of seats. Couples like quiet places with dense vegetation[54]. Finally, previous studies have shown that in addition to China, there are also different patterns of companionship in POS research in other countries[15,62,71],such as the companionship of family and friends, and the companionship of pets. Therefore, the method of understanding and designing POS from the perspective of companionship should be extended to POS of other cultures (countries).
Point 4:There is some similar research in other countries outside China - comparison would be of benefit to the paper.
Response 4:We collect some similar research cases, such as the United Kingdom, Germany, and the United States. In their research, companionship is divided into the companionship of friends, family, and pets. We made a comparison in the discussion section and borrowed some of their research conclusions. E.g:
- Previous studies have shown that the companionship of friends and pets directly affects the mental health of the elderly[61,62], which is similar to our research results.
- revious studies have already shown evidence that the companionship of pet dogs, the companionship of friends, and the companionship of family members improve physical health through physical activities so that mental health is improved[63-65].
- In the case of females, companionship gives people a sense of security during night activities because they worry about crime, violence, and harassment at night[54].
- Finally, previous studies have shown that in addition to China, there are also different patterns of companionship in POS research in other countries[15,62,71],such as the companionship of family and friends, and the companionship of pets.
We appreciate for Reviewer’s warm work earnestly and hope that the correction will meet with approval.
Once again, thank you very much for your comments and suggestions.

Reviewer 3 Report
The authors have done some interesting social research. They confirmed their assumptions about human behavior. However, it is necessary to expand the research areas to other cultures (countries) of the world.
Author Response
Dear Reviewer:
Thank you for your letter and for the reviewer’s comments concerning our manuscript entitled “Physical Activities in Public Squares: the Impact of Companionship on Chinese Residents' Health” (ID: 1270633). Those comments are all valuable and very helpful for revising and improving our paper, as well as the important guiding significance to our researches. We have studied comments carefully and have made a correction which we hope meet with approval. The main corrections in the paper and the responses to the reviewer’s comments are as flowing:
Point 1:The authors have done some interesting social research. They confirmed their assumptions about human behavior. However, it is necessary to expand the research areas to other cultures (countries) of the world.
Response 1:This is a very perfect suggestion. We have added relevant content to the discussion, especially understanding the distribution and activities of people in public spaces from the perspective of companionship, which is of great help to the design of public open spaces. Some of the public open space design strategies that we have contributed are not only applicable to China but can also be extended to different cultures (countries). The results of our discussion are presented as follows:
5.3 Some practical implications for POS design
We believe that two points should be given priority in the design of POS: In the first place, the accessibility and convenience of POS travel. Our survey respondents are mostly middle-aged and elderly people (53%), who mainly choose to travel by walk (69%). In the context of healthy aging, we must pay attention to the accessibility of the elderly to the POS and the comfort of the walking environment. Moreover, we must pay attention to social elements in the design of POS, consider the needs of companionship. For example, children like a flat and open POS where they can run and play[54]. The square dance activities of the elderly take up more space and need bright lights at night[54]. Meantime, some elderly people’s rest areas require a lot of seats. Couples like quiet places with dense vegetation[54]. Finally, previous studies have shown that in addition to China, there are also different patterns of companionship in POS research in other countries[15,62,71],such as the companionship of family and friends, and the companionship of pets. Therefore, the method of understanding and designing POS from the perspective of companionship should be extended to POS of other cultures (countries).
We appreciate for Reviewer’s warm work earnestly and hope that the correction will meet with approval.
Once again, thank you very much for your comments and suggestions.

Round 2
Reviewer 2 Report
The authors have extended their paper with adequate comments and information which are sufficient from the reviewers point of view. As usual - possibly the pepar could have been better, if the authors included photograps of the site showing the squares instead of just describing them.